# Association between different atherogenic and insulin resistance indices and infertility in polycystic ovary syndrome patients

Kasra Jafari[1,2], Nazanin Azmi-Naei[3], Nooshan Tajik[4]*, Ashraf Moini[5,6,7], Amene Abiri🄳[8]*

1 Research Development Center, Arash Women's Hospital, Tehran University of Medical Sciences, Tehran, Iran, 2 Department of Epidemiology, School of Health, Iran University of Medical Sciences, Tehran, Iran, 3 Department of Epidemiology, School of Public Health, Shahroud University of Medical Sciences, Shahroud, Iran, 4 Department of Obstetrics and Gynecology, Arash Women's Hospital, Tehran University of Medical Sciences, Tehran, Iran, 5 Department of Obstetrics and Gynecology, Endocrinology and Female Infertility Unit, Arash Women's Hospital, Tehran University of Medical Sciences, Tehran, Iran, 6 Breast Disease Research Center (BDRC), Tehran University Of Medical Sciences, Tehran, Iran, 7 Department of Endocrinology and Female Infertility, Reproductive Biomedicine Research Center, Royan Institute for Reproductive Biomedicine, ACECR, Tehran, Iran, 8 Department of Obstetrics and Gynecology, School of Medicine, Arash Women's Hospital, Tehran University of Medical Sciences, Tehran, Iran

* tajik.med321@gmail.com (NT); abiri@tums.ac.ir (AA)

## Abstract

### Background

This study aimed to assess the association between multiple atherogenic and insulin resistance indices—including TyG, TyG-BMI, AIP, TG-HDL, and METS-IR—and infertility in women with PCOS, and to evaluate their clinical utility.

### Methods

In this cross-sectional study, we analyzed data from 669 sexually active women aged 18–45 with PCOS treated at a tertiary center in Iran (2021–2023). We compared infertile women with PCOS (≥12 months unsuccessful conception, n = 274) to fertile women with PCOS (n = 395). Logistic regression, restricted cubic spline, and receiver operating characteristic (ROC) analysis were used to assess associations and predictive performance.

### Results

Infertile women had significantly worse metabolic profiles across all indices (P < 0.001). AIP showed the largest point estimate of association (adjusted OR = 9.065, 95% CI: 1.722–47.721). TG-HDL showed a significant non-linear relationship, with ORs rising steeply up to a ratio of ~2.5. Predictive ability was modest, with AUCs ranging from 0.598 (TyG-BMI) to 0.625 (AIP, TG-HDL).

**Data availability statement:** No - some restrictions will apply; In order to protect the patients' privacy and sensitive information, sharing of the data that support the findings of this study are restricted by the Tehran University of Medical Sciences ethics committee and are available from this committee (E-mail: ethics@tums.ac.ir) upon reasonable request.

**Funding:** The author(s) received no specific funding for this work.

**Competing interests:** The authors have declared that no competing interests exist.

**Abbreviations:** SD, standard deviation; PCOS, polycystic ovary syndrome; ROC, receiver operating characteristic; FBS, fasting blood sugar; HDL, High-density lipoprotein; LDL, low-density lipoprotein; BMI, Body mass index; TyG, Triglyceride-Glucose; AIP, Atherogenic Index of Plasma; TG-HDL, triglyceride-to-HDL cholesterol ratio; METS-IR, Metabolic Score for Insulin Resistance; RCS, restricted cubic spline; eGDR, Estimated glucose disposal rate; CMI, Cardiometabolic Index.

## Conclusion

Several metabolic indices, particularly AIP, are associated with infertility in women with PCOS, though their standalone predictive utility remains limited.

## Introduction

With global prevalence rates estimated between 4% and 21%, polycystic ovary syndrome (PCOS) represents one of the most common endocrine disorders in women of reproductive age [1]. Characterized by clinical features such as androgen excess, ovulatory dysfunction, and polycystic ovarian morphology, PCOS is a major contributor to female infertility, accounting for approximately 80% of anovulatory infertility cases [2].

Insulin resistance (IR) plays a well-established role in infertility and is closely linked to PCOS [3,4]. IR leads to hyperinsulinemia, which increases ovarian androgen production and disrupts folliculogenesis, resulting in anovulation [5]. It also alters the hypothalamic-pituitary-ovarian axis by raising luteinizing hormone relative to follicle-stimulating hormone, further impairing ovulatory function [6]. Additionally, IR activates the PI3K/AKT pathway in the endometrium, causing abnormal proliferation and reduced receptivity. These mechanisms collectively contribute to infertility [7]. Adipose tissue functions not only as an energy reservoir but also as a significant endocrine organ capable of producing and metabolizing sex hormones, including androgens. It serves as a site where androgens, estradiol, and dehydroepiandrosterone are converted into various estrogens, influencing reproductive function [8]. Furthermore, elevated lipid levels in the ovarian environment can disrupt oocyte maturation, potentially leading to infertility [9].

The metabolic score for insulin resistance (METS-IR) is a practical marker estimating IR using BMI, triglycerides, and fasting glucose [10]. It reflects important metabolic changes and is associated with conditions like type 2 diabetes, cardiovascular diseases, and inflammatory disorders such as psoriasis [11–13]. METS-IR is also linked to PCOS, suggesting its value as a non-invasive tool to assess metabolic risk in affected women [14]. Lipid-based markers have proven valuable in assessing insulin resistance and metabolic dysfunction in women with PCOS. Elevated triglyceride-to-HDL ratios, common in these patients, are linked to increased cardiometabolic risk [15,16]. Indices like TyG and TyG-BMI, which combine triglycerides with glucose or BMI, serve as practical tools for detecting early metabolic dysfunction, including prediabetes [17]. Considering adipose tissue's role in insulin resistance via adipokines and inflammation, these markers may enhance metabolic risk evaluation alongside METS-IR in PCOS [18].

Although several studies have explored the association between IR related indices and female infertility, most have been conducted in the general population using large survey datasets such as NHANES, without focusing on women with PCOS, a group with a distinct metabolic and reproductive profile. Moreover, prior research has

typically evaluated single indices in isolation, with limited assessment of non-linear associations or predictive performance comparison, and without considering the atherogenic index of plasma (AIP).

To address these gaps, the present study aimed to comprehensively evaluate and compare multiple atherogenic and IR indices (including TyG, TyG-BMI, AIP, TG–HDL, and METS-IR) in relation to infertility specifically among women with PCOS. In addition, we assessed potential non-linear relationships and the discriminatory performance of these indices to clarify their clinical relevance within this high-risk population.

## Materials and methods

### Study design and participants

This cross-sectional study utilized medical records of all patients diagnosed with polycystic ovary syndrome (PCOS) based on the Rotterdam 2003 criteria [19] at Arash Women's Hospital in Tehran, Iran. Data from March 1st, 2021, to December 31st, 2023, were extracted for analysis (all data extracted on February 20th, 2024). The authors did not have access to information that could identify individual participants during or after data collection

### Eligibility criteria

The study included sexually active women aged 18–45 diagnosed with PCOS. Exclusion criteria were: (1) non-sexually active women, (2) those with current or previous cancer diagnoses, and (3) cases with male factor infertility (as recorded in medical files).

### Measurements and tools

Blood samples were collected after an overnight fast of at least 8–12 hours. Serum levels of TG, HDL, LDL, and FBS were measured using standard methods with commercially available kits (Pars Azmoon, Tehran, Iran), according to the manufacturer's instructions and routine laboratory quality-control procedures.

Anthropometric measurements were obtained by trained clinical staff. Body weight was measured using a Seca 813 digital scale (accurate to 50 g) with participants wearing light clothing and no shoes, and height was measured using a wall-mounted stadiometer.

Infertility status was determined based on documentation in medical records, defined as failure to achieve a clinical pregnancy after at least 12 months of regular unprotected intercourse. This information was recorded during routine clinical evaluations by treating gynecologists and was not based solely on self-report. Women with documented male factor infertility were excluded. Other major female causes of infertility unrelated to PCOS (such as known uterine anomalies, tubal obstruction, or premature ovarian insufficiency) were assessed when available in the medical records; however, infertility was classified as all-cause infertility within a PCOS population.

PCOS was diagnosed by experienced gynecologists based on the revised 2003 Rotterdam criteria. According to these criteria, PCOS was defined by the presence of at least two of the following three features: (1) oligo- or anovulation, (2) clinical and/or biochemical signs of hyperandrogenism, and (3) polycystic ovarian morphology on ultrasound, after exclusion of other etiologies.

Several metabolic indices were calculated using fasting laboratory values and anthropometric measurements.

Body mass index (BMI) was calculated as weight in kilograms divided by height in meters squared:

$$BMI = \frac{weight}{height^2}$$

The Triglyceride-Glucose (TyG) index was calculated as the natural logarithm of the product of fasting triglycerides and fasting blood glucose divided by two:

$$TyG = \ln\left(\frac{FBS \times TG}{2}\right)$$

To further enhance the metabolic predictive capacity, the TyG-BMI index was calculated by multiplying the TyG index by the BMI:

$$TyG - BMI = TyG \times BMI$$

The AIP was determined by taking the base-10 logarithm of the ratio of triglycerides to HDL cholesterol:

$$AIP = \log_{10}\left(\frac{TG}{HDL}\right)$$

The triglyceride-to-HDL cholesterol ratio (TG-HDL) was calculated by dividing the serum triglyceride level by HDL cholesterol:

$$TG - HDL = \left(\frac{TG}{HDL}\right)$$

Finally, the Metabolic Score for Insulin Resistance (METS-IR) was calculated using the following validated formula:

$$METS - IR = \frac{\ln\left((2 \times FBS) + TG\right) \times BMI}{\ln(HDL)}$$

### Ethical considerations

This study received ethical approval from the Tehran University of Medical Sciences ethics committee (IR.TUMS.MEDICINE.REC.1402.582, approval date: 2024-01-21). All procedures complied with Iran's national guidelines for medical research.

### Statistical analysis

Continuous variables are presented as mean ± standard deviation (SD) and were compared using independent t-tests. Categorical variables were analyzed using chi-square tests. We evaluated associations using three sequential logistic regression models: (1) an unadjusted model, (2) a model adjusted for age, and (3) a model adjusted for age, BMI, and LDL, with covariate selection informed by a literature review. For linear associations, we used standard binary logistic regression, while non-linear relationships were modeled using restricted cubic splines (RCSs) with four knots. We assessed the predictive performance of each atherogenic index through receiver operating characteristic (ROC) curve analysis. All analyses were performed in R version 4.5.0, with statistical significance defined as $\alpha = 0.05$ (two-tailed).

## Results

### Participants characteristics

Of the initially 1,004 assessed women, 335 women were excluded according to the exclusion criteria, and 669 women were included in the final analysis, of whom 274 (40.96%) were classified as infertile and 395 (59.04%) as fertile. The mean age of participants was 32.33 ± 5.69 years, with infertile women being significantly older than fertile women (33.11 ± 5.66 vs. 31.79 ± 5.65 years, $P = 0.003$).

Anthropometric and metabolic parameters showed significant differences between the two groups. Infertile women had a significantly higher body mass index (BMI) compared to fertile women (26.02±4.95 vs. 24.59±4.19 kg/m², $P<0.001$).

Although serum Anti-Müllerian Hormone (AMH) levels were lower in infertile women (4.58±3.89 ng/mL) compared to fertile women (5.11±4.68 ng/mL), this difference was not statistically significant ($P=0.122$). Serum vitamin D levels were similar between groups (28.50±14.97 vs. 28.75±14.25 ng/mL, $P=0.829$).

Regarding lipid profiles and glucose metabolism, infertile women had significantly higher levels of triglycerides (123.60±43.20 vs. 108.95±41.99 mg/dL, $P<0.001$), low-density lipoprotein (LDL) cholesterol (105.35±29.44 vs. 98.03±58.08 mg/dL, $P<0.001$), and fasting blood sugar (FBS) (99.32±36.98 vs. 92.09±9.76 mg/dL, $P<0.001$). Conversely, high-density lipoprotein (HDL) cholesterol levels were lower in the infertile group (40.09±5.10 vs. 42.13±4.71 mg/dL, $P<0.001$).

Infertile women also demonstrated significantly higher values for all examined atherogenic and insulin resistance indices, including the TyG (8.64±0.45 vs. 8.45±0.40, $P<0.001$), TyG-BMI (226.45±54.48 vs. 209.24±44.80, $P<0.001$), AIP (0.47±0.19 vs. 0.39±0.19, $P<0.001$), TG-HDL ratio (3.25±1.48 vs. 2.72±1.40, $P<0.001$), and METS-IR (41.09±10.55 vs. 37.68±8.53, $P<0.001$). Other demographic and clinical characteristics of the study participants are presented in Table 1.

### Infertility and atherogenic indices

In the unadjusted model (Model 1), all indices demonstrated significant positive associations with infertility (Table 2). The TyG index showed a strong association (OR = 2.787, 95% CI: 1.914–4.057, $P<0.001$), which remained robust after adjusting for age in Model 2 (OR = 2.716, 95% CI: 1.859–3.967, $P<0.001$) and after further adjustment for LDL in Model 3 (OR = 2.901, 95% CI: 1.393–6.040, $P=0.004$).

The TyG-BMI index was also significantly associated with infertility in the unadjusted (OR = 1.007, 95% CI: 1.004–1.011, $P<0.001$) and age-adjusted model (OR = 1.007, 95% CI: 1.004–1.011, $P<0.001$), also in the Model 3 (OR = 1.030, 95% CI: 1.003–1.059, $P=0.028$).

The AIP showed the largest point estimate of association among all indices. In the unadjusted model, the odds of infertility increased more than eightfold with each unit increase in AIP (OR = 8.677, 95% CI: 3.833–19.640, $P<0.001$). This association remained highly significant in both the age-adjusted model (OR = 8.440, 95% CI: 3.710–19.196, $P<0.001$) and the fully adjusted model (OR = 9.065, 95% CI: 1.722–47.721, $P=0.009$).

The TG-HDL ratio was significantly associated with infertility in Models 1 and 2 (OR = 1.294, 95% CI: 1.159–1.444, $P<0.001$ and OR = 1.286, 95% CI: 1.151–1.436, $P<0.001$, respectively), but the association weakened and lost significance in the fully adjusted model (OR = 1.164, 95% CI: 0.947–1.432, $P=0.148$).

Similarly, METS-IR was significantly associated with infertility in Models 1 and 2 (both OR = 1.039, 95% CI: 1.021–1.057, $P<0.001$). After adjustment for LDL in Model 3, the association was also statistically significant (OR = 1.162, 95% CI: 1.031–1.309, $P=0.014$).

An RCS model was used to assess the non-linear relationship between the TG-HDL ratio and the odds of infertility among women with PCOS (Fig 1). The overall association was statistically significant (p<0.001), with a significant non-linear component (p=0.011). The curve illustrates a steep increase in the odds of infertility with rising TG-HDL values up to approximately 2.5. However, beyond a TG-HDL value of approximately 2.5, the 95% CI crosses the red dashed line (representing OR = 1), indicating that the association is no longer statistically significant in that range.

### Predictive performance

The predictive abilities of the evaluated metabolic indices for identifying infertility among women with PCOS were assessed using receiver operating characteristic (ROC) curve analysis. As illustrated in Fig 2, the area under the curve (AUC) values were highest for the AIP and the TG-HDL, both with an AUC of 0.625, indicating modest discriminatory power. The TyG index also demonstrated a comparable performance with an AUC of 0.624. The METS-IR score yielded a slightly lower AUC of 0.601, while the TyG-BMI index had the lowest predictive value among the evaluated indices, with an AUC of 0.598.

**Table 1. Demographic and clinical characteristics of the study participants.**

| Characteristics | | Total | Fertile | Infertile | P-value |
|---|---|---|---|---|---|
| | | N = 669 | N = 395 | N = 274 | |
| Age, years (SD) | | 32.33 (5.69) | 31.79 (5.65) | 33.11 (5.66) | **0.003** |
| AMH, ng/dL (SD) | | 4.89 (4.38) | 5.11 (4.68) | 4.58 (3.89) | 0.122 |
| BMI, kg/m² (SD) | | 25.17 (4.57) | 24.59 (4.19) | 26.02 (4.95) | **<0.001** |
| Vitamin D, ng/dL (SD) | | 28.65 (14.54) | 28.75 (14.25) | 28.50 (14.97) | 0.829 |
| TG, mg/dL (SD) | | 114.95 (43.07) | 108.95 (41.99) | 123.60 (43.20) | **<0.001** |
| HDL, mg/dL (SD) | | 41.29 (4.97) | 42.13 (4.71) | 40.09 (5.10) | **<0.001** |
| LDL, mg/dL (SD) | | 101.03 (48.54) | 98.03 (58.08) | 105.35 (29.44) | **<0.001** |
| FBS, mg/dL (SD) | | 95.05 (25.05) | 92.09 (9.76) | 99.32 (36.98) | **<0.001** |
| TyG (SD) | | 8.53 (0.43) | 8.45 (0.40) | 8.64 (0.45) | **<0.001** |
| TyG-BMI (SD) | | 216.29 (49.68) | 209.24 (44.80) | 226.45 (54.48) | **<0.001** |
| AIP (SD) | | 0.42 (0.19) | 0.39 (0.19) | 0.47 (0.19) | **<0.001** |
| TG-HDL (SD) | | 2.94 (1.46) | 2.72 (1.40) | 3.25 (1.48) | **<0.001** |
| METS-IR (SD) | | 39.07 (9.55) | 37.68 (8.53) | 41.09 (10.55) | **<0.001** |
| Tobacco consumption (%) | No | 568 (84.90%) | 335 (84.81%) | 233 (85.04%) | 0.936 |
| | Yes | 101 (15.10%) | 60 (15.19%) | 41 (14.96%) | |
| Acne (%) | No | 447 (66.82%) | 257 (65.06%) | 190 (69.34%) | 0.248 |
| | Yes | 222 (33.18%) | 138 (34.94%) | 84 (30.66%) | |
| Alopecia (current or former) (%) | No | 508 (75.93%) | 292 (73.92%) | 216 (78.83%) | 0.144 |
| | Yes | 161 (24.07%) | 103 (26.08%) | 58 (21.17%) | |
| Hirsutism (current or former) (%) | No | 340 (50.82%) | 201 (50.89%) | 139 (50.73%) | 0.968 |
| | Yes | 329 (49.18%) | 194 (49.11%) | 135 (49.27%) | |
| Oligomenorreha (%) | No | 184 (27.50%) | 114 (28.86%) | 70 (25.55%) | 0.345 |
| | Yes | 485 (72.50%) | 281 (71.14%) | 204 (74.45%) | |
| Polymenorreha (%) | No | 494 (73.84%) | 289 (73.16%) | 205 (74.82%) | 0.632 |
| | Yes | 175 (26.16%) | 106 (26.84%) | 69 (25.18%) | |
| Hypermenorreha (%) | No | 484 (72.35%) | 295 (74.68%) | 189 (68.98%) | 0.105 |
| | Yes | 185 (27.65%) | 100 (25.32%) | 85 (31.02%) | |
| Dysmenorreha (%) | No | 340 (50.82%) | 221 (55.95%) | 119 (43.43%) | **0.001** |
| | Yes | 329 (49.18%) | 174 (44.05%) | 155 (56.57%) | |
| Dyspareunia (%) | No | 479 (71.60%) | 306 (77.47%) | 173 (63.14%) | **<0.001** |
| | Yes | 190 (28.40%) | 89 (22.53%) | 101 (36.86%) | |

Acronyms: BMI: Body Mass Index, AMH: Anti-Müllerian Hormone, TyG: Triglyceride-Glucose Index, TyG-BMI: Triglyceride-Glucose Body Mass Index, AIP: Atherogenic Index of Plasma, TG-HDL: triglyceride to High-density lipoprotein ratio, METS-IR: Metabolic Score for Insulin Resistance.

The AUC values observed for all indices ranged from 0.598 to 0.625, indicating poor to modest discriminative ability. These findings suggest that, when used individually, the evaluated metabolic indices have limited capacity to distinguish between infertile and fertile women with PCOS.

## Discussion

### Principal findings

In this cross-sectional study of 669 women with polycystic ovary syndrome (PCOS), we evaluated the association and predictive performance of several metabolic indices—including TyG, TyG-BMI, AIP, TG-HDL, and METS-IR—in relation to infertility. All indices were significantly higher in infertile women compared to their fertile counterparts and

**Table 2. Association between infertility and different atherogenic indices.**

| Index | Model 1[a] | | Model 2[b] | | Model 3[c] | |
|---|---|---|---|---|---|---|
| | Odds ratio (95% confidence interval) | P-value | Odds ratio (95% confidence interval) | P-value | Odds ratio (95% confidence interval) | P-value |
| **TyG** | 2.787(1.914-4.057) | < 0.001 | 2.716(1.859-3.967) | < 0.001 | 2.901(1.393-6.040) | 0.004 |
| **TyG-BMI** | 1.007(1.004-1.011) | < 0.001 | 1.007(1.004-1.011) | < 0.001 | 1.030(1.003-1.059) | 0.028 |
| **AIP** | 8.677(3.833-19.640) | < 0.001 | 8.440(3.710-19.196) | < 0.001 | 9.065(1.722-47.721) | 0.009 |
| **TG-HDL** | 1.294(1.159-1.444) | < 0.001 | 1.286(1.151-1.436) | < 0.001 | 1.164(0.947-1.432) | 0.148 |
| **METS-IR** | 1.039(1.021-1.057) | < 0.001 | 1.039(1.021-1.057) | < 0.001 | 1.162(1.031-1.309) | 0.014 |

Acronyms: TyG: Triglyceride-Glucose Index, TyG-BMI: triglyceride glucose body mass index, AIP: Atherogenic Index of Plasma, TG-HDL: triglyceride to High-density lipoprotein ratio, METS-IR: Metabolic Score for Insulin Resistance.
a. Unadjusted model.
b. Adjusted for age.
c. Adjusted for age, body mass index (BMI), and Low-density lipoprotein (LDL).

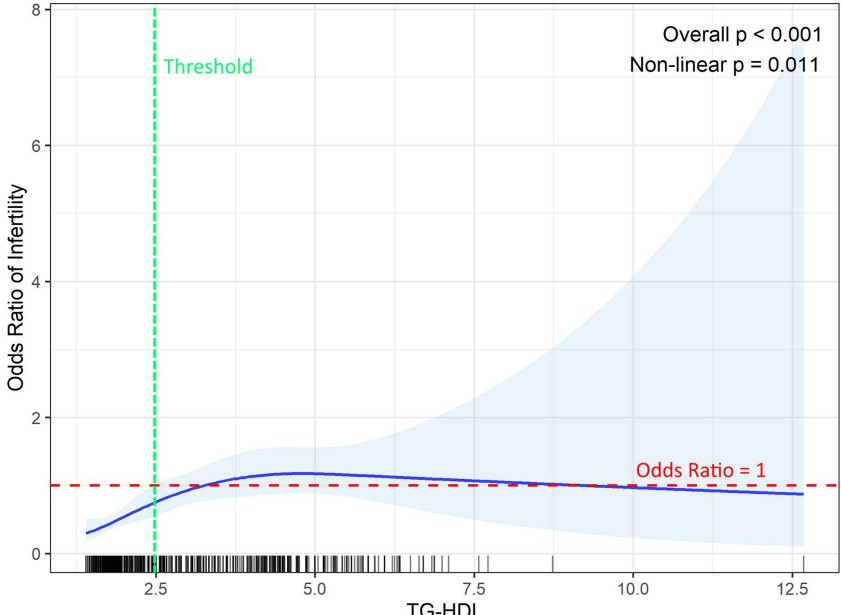

**Fig 1. Restricted cubic spline fitting for the association between TG-HDL ratio and female infertility.** The red dashed horizontal line represents an odds ratio (OR) of 1, indicating no association. The green dashed vertical line indicates the threshold (approximately = 2.5) where the association loses statistical significance. The barcode plot on the X axis shows the sample size density across the TG-HDL ratio range.

showed positive associations with infertility in unadjusted, age-adjusted, and fully adjusted logistic regression models. Notably, AIP demonstrated the largest point estimate of association, with an odds ratio exceeding 8, even after full adjustment for confounders. Despite these statistically significant associations, the predictive performance of all indices was modest, with AUC values ranging from 0.598 to 0.625. These results indicate that, although metabolically informative, none of the evaluated indices are suitable as standalone tools for clinical prediction or screening of infertility in women with PCOS.

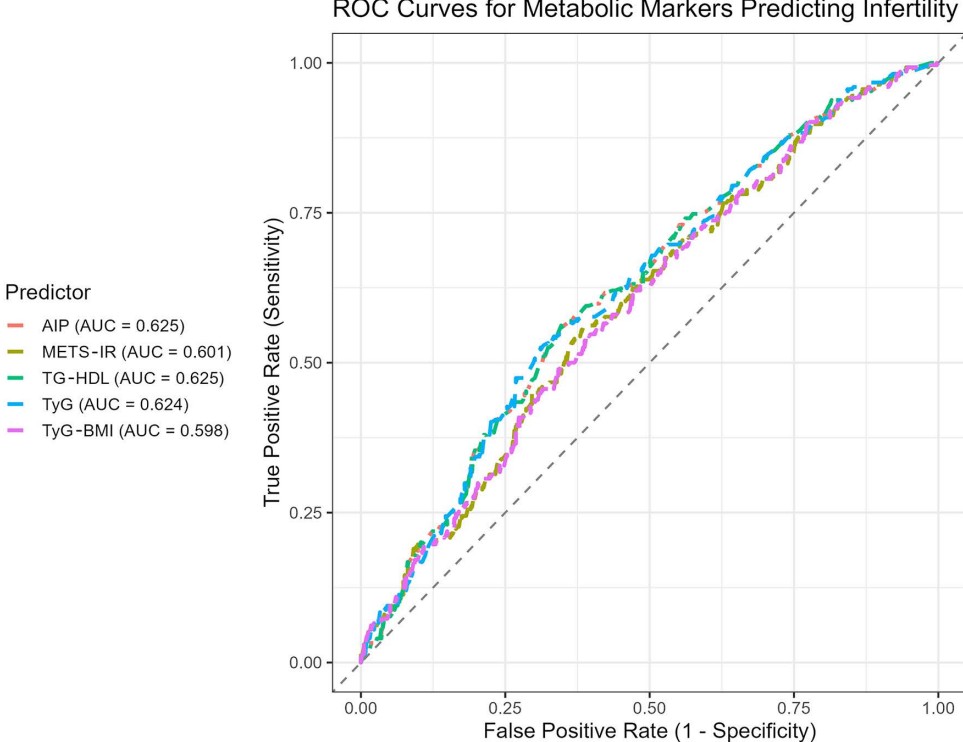

**Fig 2. ROC curves for atherogenic indices to predict female infertility.**

## Biological pathways

IR can lead to several problems that affect fertility in women with PCOS. IR causes high levels of insulin, which can disturb the balance of hormones needed for normal ovulation [20,21]. In addition to this, IR increases inflammation in the body by raising levels of cytokines such as TNF-α and IL-6. These inflammatory substances can negatively affect how the ovaries work and reduce the quality of the eggs [22]. IR can also cause oxidative stress, which damages the cells in the ovaries and makes it harder for good-quality eggs to develop [23]. These problems, together with hormonal imbalance, can make it more difficult for women with PCOS to become pregnant.

Fat tissue also plays an important role in this process. Besides storing energy, fat tissue produces hormones that affect reproduction. In PCOS, too much fat—especially around the abdomen—can change the levels of hormones like leptin and adiponectin. High leptin levels and low adiponectin levels can both interfere with insulin sensitivity and hormone regulation [24]. This creates a cycle that worsens both metabolic and reproductive problems [21]. Also, high levels of fats in the ovaries can damage the cells that support egg development, leading to poor egg quality [25]. These biological changes may help explain the link between atherogenic and insulin resistance indices and infertility in PCOS.

## Results in the context of what is known

Our findings are broadly consistent with previous research exploring the association between metabolic indices and female infertility, though our study adds to the literature by focusing specifically on women with PCOS and evaluating a more comprehensive range of indices. Li et al. (2025), analyzing data from the NHANES study, found a significant

association between the METS-IR and infertility, with adjusted odds ratios (aORs) consistently at 1.02 (95% CI: 1.01–1.04), even after controlling for a wide range of sociodemographic and clinical confounders [26]. Our study also identified a significant association between METS-IR and infertility in unadjusted and age-adjusted models; additionally, the association kept its significance after further adjustment for BMI and LDL.

Similarly, Liu et al. (2025) and Kong et al. (2024) demonstrated significant associations between the Cardiometabolic Index (CMI) and infertility, with adjusted ORs of 1.27 and 1.12, respectively, though predictive performance was not reported [27,28]. While our study did not evaluate CMI specifically, we included closely related indices such as TG-HDL and AIP, which similarly reflect lipid-glucose interactions. Zhuang et al. (2024) assessed the TyG index and reported an adjusted OR of 1.51 (95% CI: 1.14–2.00) with a modest AUC of 0.56 [29]. Our findings confirm the association of TyG with infertility, showing a stronger effect size (fully adjusted OR = 2.901) and a higher AUC of 0.624, though still in the modest range. The greater strength of association in our study may reflect the higher metabolic burden of PCOS patients compared to the general population examined in NHANES studies.

Li et al. (2024) focused on the estimated glucose disposal rate (eGDR), an insulin sensitivity index, and found a significant inverse association with infertility (adjusted OR = 0.86), along with an AUC of 0.632 [30]. While we did not assess eGDR, our findings on METS-IR—a related insulin resistance marker—are directionally consistent, albeit with slightly lower predictive value (AUC = 0.601). Notably, Xia et al. (2023) investigated both TyG and TyG-BMI indices, reporting adjusted ORs of 1.37 and 1.01, respectively, and AUCs of 0.62 and 0.68 [31]. In our PCOS participants, we observed similar associations for both indices, though TyG-BMI had the lowest AUC (0.598), suggesting that the added value of BMI in TyG-BMI may be limited in populations already at high risk for metabolic dysfunction.

Importantly, none of the reviewed studies included the AIP, which in our analysis emerged as the strongest predictor of infertility, with an adjusted OR of 9.065 (95% CI: 1.722–47.721) and an AUC of 0.625. This highlights the novelty of our study in evaluating AIP in the context of reproductive outcomes. However, consistent with previous literature, the overall predictive performance of all indices—ranging from AUCs of 0.598 to 0.625—was modest, underscoring the limited clinical utility of these markers when used in isolation. Table 3- Findings of studies for the association between different atherogenic and insulin resistance indices and female infertility.

## Clinical implications

Although the discriminative performance of the metabolic indices examined in this study was modest, our findings suggest that unfavorable lipid-related profiles may help identify subgroups of women with PCOS who are at increased risk of infertility. These indices are readily available in routine clinical practice and may serve as adjunctive markers for risk stratification and early counseling rather than as standalone predictive tools. From a clinical perspective, recognition of adverse metabolic patterns in women with PCOS may support timely lifestyle interventions as part of a more general fertility-focused management strategy. However, these indices should not be used in isolation for clinical prediction, and their utility should be confirmed in prospective studies.

## Strengths and limitations

This study has several notable strengths. First, it is among the few to comprehensively assess multiple atherogenic and insulin resistance indices—including AIP, TyG, TyG-BMI, TG-HDL, and METS-IR—in relation to infertility specifically within a PCOS population, rather than the general population. Given the metabolic vulnerability of women with PCOS, this targeted approach enhances the clinical relevance of our findings. Second, the use of a relatively large and well-defined sample (n = 669) from a tertiary care center adds strength and improves the precision of our estimates. Moreover, we assessed predictive performance using ROC curve analysis, offering insight into the clinical utility of these indices.

However, our study also has limitations. Its cross-sectional design precludes causal inference between metabolic indices and infertility. Although infertility status was clinically documented, a detailed etiologic workup was not uniformly

**Table 3. Findings of studies for the association between different atherogenic and insulin resistance indices and female infertility.**

| Author (year) | Population | Investigated Indices | Association | Predictive Performance |
|---|---|---|---|---|
| Li et al. (2025) [26] | 1,541 participants from the National Health and Nutrition Examination Survey (NHANES) from 2013 to 2018. Those aged over 45 or under 18 years, and those with missing data on METS-IR or infertility were excluded. | METS-IR | • Unadjusted: OR = 1.02 (1.01–1.04)*** <br> • Adjusted for age and ethnicity: OR = 1.02 (1.01–1.03)** <br> • Further adjusted for education, poverty-income ratio, marital status, drinking, blood cotinine, dyslipidemia, diabetes, hypertenstion, mesntural status, pelvic inflammatory disease, contraceptive pill use, female hormone use: OR = 1.02 (1.01–1.04)** | Not disclosed |
| Liu et al. (2025) [27] | 3,613 participants from the NHANES database from 2013 to 2020. Individuals aged < 20 years or > 45 years, lacked information on fertility status, and incomplete CMI measurements were excluded. | CMI | • Adjusted for age, race, education: OR = 1.34 (1.16–1.55)*** <br> • Adjusted for age, race, education, ratio of income; marital status, smoke; hypertension; diabetes: OR = 1.27 (1.08–1.49)** | Not disclosed |
| Zhuang et al. (2024) [29] | 1,556 participants from the NHANES database from 2013 to 2020. Participants lacking data on TyG index, fertility information, and participants older than 45 years or younger than 18 years were excluded. | TyG | • Unadjusted: OR = 1.41 (1.12–1.79)** <br> • Adjusted for age and race: OR = 1.45 (1.14–1.85)** <br> • Adjusted for age, ratio of family income to poverty, race, education level, marital status, smoked at least 100 cigarettes, had at least 12 alcohol drinks/1 year, ever treated for a pelvic infection/PID, ever taken birth control pills, had regular periods in past 12 months: OR = 1.51 (1.14–2.00)** | AUC = 0.56 |
| Li et al. (2024) [30] | 2,541 participants from the NHANES database from 2013 to 2020. Participants aged < 18 or > 45 years, without infertility questionnaire data, without complete eGDR data, and missing data on covariates information were excluded. | eGDR | • Unadjusted: OR = 0.85 (0.80–0.90)*** <br> • Adjusted for age, race/ethnicity, and education: OR = 0.87 (0.81–0.93)*** <br> • Adjusted for age, race/ethnicity, education, poverty income ratio, marital status, smoking, drinking, hemoglobin, total cholesterol, diabetes, regular period and pelvic inflammatory disease, adrenocortical insufficiency and sex hormone dysfunction: OR = 0.86 (0.80–0.94)*** | AUC = 0.632 |
| Kong et al. (2024) [28] | 1,720 participants from the NHANES database from 2013 to 2020. Minor (age < 18) participants excluded, and people with full CMI and infertility data were picked for inclusion. | CMI | • Unadjusted: OR = 1.10 (1.01–1.21)* <br> • Adjusted for age and race: OR = 1.10 (1.01–1.21)* <br> • Adjusted for age, race, ratio of family income to poverty, education level, marital status, diabetes, cardiovascular disease history, smoked at least 100 cigarettes, age when first menstrual period occurred, ever taken birth control pills, ever treated for a pelvic infection/pelvic inflammatory disease, had regular periods in past 12 months: OR = 1.12 (1.01–1.24)* | Not disclosed |
| Xia et al. (2023) [31] | 1,043 participants from the NHANES database from 2013 to 2020. Participants aged < 18 or > 36 years, without infertility data, with missing BMI, triglyceride, serum insulin and fasting glucose data, and missing data were excluded. | TyG and TyG-BMI | TyG: <br> • Unadjusted: OR = 2.00 (1.43–2.78)*** <br> • Adjusted for age: OR = 1.80 (1.29–2.52)*** <br> • Adjusted for age, marital status, education, LDL, hypertension, diabetes: OR = 1.37 (0.93–2.01) <br> TyG-BMI: <br> • Unadjusted: OR = 1.01 (1.00–1.01)*** <br> • Adjusted for age: OR = 1.01 (1.00–1.01)*** <br> • Adjusted for age, marital status, education, LDL, hypertension, diabetes: OR = 1.01 (1.00–1.01)*** | AUC TyG = 0.62 <br> AUC TyG-BMI = 0.68 |

Acronyms: BMI: Body Mass Index, TyG: Triglyceride-Glucose Index, TyG-BMI: triglyceride glucose body mass index, METS-IR: Metabolic Score for Insulin Resistance, eGDR: Estimated glucose disposal rate, CMI: Cardiometabolic Index.

* P-value < 0.05, ** P-value < 0.01, *** P-value < 0.001.

available for all participants; therefore, residual misclassification of infertility causes cannot be fully excluded. Moreover, important potential confounders such as dietary patterns, physical activity, and other lifestyle factors were not available in the medical records and therefore could not be adjusted for. Similarly, detailed reproductive factors were not comprehensively available to assess. Residual confounding from these unmeasured factors may have influenced the observed associations. Also, selection bias may be present, as the study population was drawn from a tertiary care center, which may overrepresent women with more severe metabolic or reproductive manifestations of PCOS. Finally, while we evaluated several indices, some newer or less commonly used markers (e.g., eGDR, CMI) were not included, which may limit comparability with all prior research.

## Conclusion

In this cross-sectional study of women with polycystic ovary syndrome, we found that several metabolic indices—including AIP, TyG, TyG-BMI, TG-HDL, and METS-IR—were significantly associated with infertility, with AIP demonstrating the largest point estimate of association. Although several metabolic indices—particularly AIP—were independently associated with infertility, their discriminatory performance was limited. Therefore, these indices should not be used as standalone clinical predictors. Future research should focus on integrated, multifactorial models combining metabolic markers with reproductive hormones, ovulatory function, and lifestyle factors. Prospective cohort studies are also needed to establish temporal and causal relationships between metabolic health and infertility.

## Acknowledgments

Thanks to the Arash Women's Hospital staff for their help.

## Author contributions

**Conceptualization:** Kasra Jafari, Nazanin Azmi-Naei, Nooshan Tajik, Ashraf Moini, Amene Abiri.

**Data curation:** Kasra Jafari.

**Formal analysis:** Kasra Jafari.

**Investigation:** Nooshan Tajik, Ashraf Moini, Amene Abiri.

**Methodology:** Kasra Jafari, Nazanin Azmi-Naei, Nooshan Tajik, Ashraf Moini, Amene Abiri.

**Project administration:** Nooshan Tajik, Amene Abiri.

**Resources:** Nooshan Tajik, Ashraf Moini.

**Supervision:** Amene Abiri.

**Visualization:** Kasra Jafari, Nazanin Azmi-Naei.

**Writing – original draft:** Kasra Jafari, Nazanin Azmi-Naei, Nooshan Tajik, Ashraf Moini, Amene Abiri.

**Writing – review & editing:** Kasra Jafari, Nazanin Azmi-Naei, Nooshan Tajik, Ashraf Moini, Amene Abiri.

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
