## [Decision Letter · Decision Letter 0]

19 Dec 2025

PONE-D-25-55841Association between different atherogenic and insulin resistance indices and infertility in polycystic ovary syndrome patientsPLOS One

Dear Dr. Abiri,

Thank you for submitting your manuscript to PLOS ONE. After careful consideration, we feel that it has merit but does not fully meet PLOS ONE’s publication criteria as it currently stands. Therefore, we invite you to submit a revised version of the manuscript that addresses the points raised during the review process.

Dear authors, as you can see, the reviewers have requested substantial revisions to your manuscript. We are certainly willing to reconsider a revised submission, but please know that this is not preliminary acceptance of your paper. When returning your revised manuscript, please be sure to include a point-by-point summary of the suggestions of the reviewers that specifies how and where in the text you have addressed the suggestions.

We look forward to receiving your revised manuscript.

Kind regards,

Ricardo Ney Oliveira Cobucci, Ph.D

Academic Editor

PLOS One

Journal Requirements:

2. In the online submission form you indicate that your data is not available for proprietary reasons and have provided a contact point for accessing this data. Please note that your current contact point is a co-author on this manuscript. According to our Data Policy, the contact point must not be an author on the manuscript and must be an institutional contact, ideally not an individual. Please revise your data statement to a non-author institutional point of contact, such as a data access or ethics committee, and send this to us via return email. Please also include contact information for the third party organization, and please include the full citation of where the data can be found.

Reviewers' comments:

Reviewer's Responses to Questions

**Comments to the Author**

1. Is the manuscript technically sound, and do the data support the conclusions?

Reviewer #1: Partly

Reviewer #2: Yes

2. Has the statistical analysis been performed appropriately and rigorously? 

Reviewer #1: Yes

Reviewer #2: Yes

3. Have the authors made all data underlying the findings in their manuscript fully available?

Reviewer #1: No

Reviewer #2: Yes

4. Is the manuscript presented in an intelligible fashion and written in standard English?

Reviewer #1: Yes

Reviewer #2: No

5. Review Comments to the Author

Reviewer #1: This cross-sectional study examines the association between five metabolic indices (TyG, TyG-BMI, AIP, TG-HDL, METS-IR) and infertility in 669 Iranian women with PCOS. The authors report that all indices were significantly higher in infertile women, with AIP showing the strongest association (adjusted OR = 9.065). However, ROC analysis revealed modest predictive performance (AUCs 0.598–0.625), suggesting limited standalone clinical utility.

The study addresses a relevant clinical question and is methodologically sound in many respects. However, several issues must be addressed:

1. TyG-BMI formula in the Methods  it should be TyGxBMI not TgXBMI.

2. Specify how “unsuccessful conception” was documented and whether other causes of infertility were considered.

3. Given the modest AUCs, emphasize that these indices are not yet suitable for standalone clinical prediction.

4. Expand limitations: Address potential unmeasured confounders (e.g., diet, physical activity, detailed infertility workup).

5. Consider adding a sensitivity analysis: Excluding women with very high TG-HDL ratios (>2.5) where the association became non-significant.

Figure 1:

1. Clarity of Red Dashed Line: The caption should explicitly state that the red dashed line represents OR = 1 (no association).

2. Threshold Annotation: Adding a vertical dotted line at TG-HDL ≈ 2.5 with a brief note (e.g., “threshold”) would help readers quickly interpret the inflection point.

Figure 2:

1. The legend lists “TG_HDL” and “TyG_HDL.” However“TyG_HDL” is not described in the text.

2. Some lines are very close together (AIP, TG-HDL). Consider using dashed/dotted styles or clearer color differentiation.

3. It would be helpful to add AUC values in the legend (e.g., “AIP (AUC = 0.625)”) for immediate comparison.

4. “BMI” is included in the ROC but not discussed in the predictive performance results. Either explain its inclusion or remove it to avoid distraction.

Reviewer #2: PONE-D-25-55841

I am writing to express my opininion on Association between different atherogenic and insulin resistance indices and infertility in polycystic ovary syndrome patients which submitted in plos one journal.

This study suffred from lack of novelity, its not sutible for publication untill major revision.

Authors failed to picture the rationality of this study? What is main novelity? Previous study answer this question how you fill the gap?

How many women you excluded due to the exclusion criteria?

How were the techniues of assessments , kits, methodes…

How was diagnostic criteria of PCOS?

What is definition of infertility?

It was better to have a control group for comparison.

Please discuss about any pollsible bias of study.

What is clinical implication of study?

Editor advised to check the usig of AI.

6. PLOS authors have the option to publish the peer review history of their article (what does this mean?). If published, this will include your full peer review and any attached files.

Reviewer #1: No

Reviewer #2: No

---

## [Author Response · Author response to Decision Letter 1]

31 Jan 2026

Dear Dr. Ricardo Ney Oliveira Cobucci,

We sincerely appreciate the time and effort you and the reviewer have dedicated to evaluating our manuscript, "Association between different atherogenic and insulin resistance indices and infertility in polycystic ovary syndrome patients" (Manuscript ID: PONE-D-25-55841). We are grateful for the constructive feedback, which has helped us improve the quality of our work, and accept our sincerest apologies for the late response, since, because of the total internet shutdown in Iran, we were not able to communicate.

We have carefully addressed all the comments and suggestions provided by the reviewer. Below, we provide a point-by-point response to each remark, detailing the revisions made in the revised manuscript. All changes have been highlighted in the revised version for easy reference in yellow.

Reviewer #1

We are truly grateful for your kind and encouraging words. Your positive feedback means a great deal to our team and reinforces our commitment to further research in this area. Thank you for taking the time to review our work so thoughtfully.

Comment 1: "TyG-BMI formula in the Methods  it should be TyGxBMI not TgxBMI.”

Response 1: We corrected this typo.

Comment 2: "Specify how “unsuccessful conception” was documented and whether other causes of infertility were considered.”

Response 2: Infertility status was determined based on documentation in medical records and defined as failure to achieve pregnancy after at least 12 months of regular unprotected intercourse, as recorded during routine clinical assessments by treating gynecologists. Classification was therefore not based solely on self-report.

Male factor infertility was explicitly excluded. Other female causes of infertility unrelated to PCOS were considered when information was available in medical records; however, infertility was classified as all-cause infertility within a PCOS population. We have clarified this point in the Measurements and Tools section (lines 110-117) and acknowledged the potential for residual misclassification in the Limitations section (lines 343-344).

Comment 3: "Given the modest AUCs, emphasize that these indices are not yet suitable for standalone clinical prediction.”

Response 3: We agree with the reviewer and thank them for this valuable suggestion. We have revised the Results (lines 243-246), Discussion (lines 258-262), and Conclusion (lines 359-361) sections to explicitly state that these indices are not suitable as standalone tools for infertility prediction in women with PCOS and should be interpreted as adjunctive metabolic markers rather than independent screening instruments.

Comment 4: "Expand limitations: Address potential unmeasured confounders (e.g., diet, physical activity, detailed infertility workup).”

Response 4: We have expanded the Strengths and Limitations section (lines 344-348) to explicitly acknowledge the potential influence of unmeasured confounders (as you asked for diet, physical activity, and detailed infertility workup) and their possible role in the residual confounding.

Comment 5: "Consider adding a sensitivity analysis: Excluding women with very high TG-HDL ratios (>2.5) where the association became non-significant.”

Response 5: We thank the reviewer for this insightful suggestion. To address potential non-linearity and the influence of extreme values, we modeled the association between the TG–HDL ratio and infertility using restricted cubic spline analysis, which provides a flexible and statistically appropriate assessment without imposing arbitrary cut points. The spline analysis demonstrated a significant overall and non-linear association, with loss of statistical significance at higher TG–HDL values.

Importantly, as illustrated in the spline plot (barcode plot on the X axis of Fig. 1), the number of observations decreases substantially at TG–HDL ratios above approximately 2.5, which may partly explain the widening confidence intervals and loss of statistical significance in this range, rather than a true absence of association.

Given that non-linearity has already been formally evaluated and that post hoc exclusion based on a threshold may introduce selection bias, we chose not to perform an additional sensitivity analysis at this stage. However, if the reviewer or editor considers such an analysis essential, we would be pleased to perform it and report the results in a subsequent round of revisions or as supplementary material.

Comments on Figure 1:

Comment 1: Clarity of Red Dashed Line: The caption should explicitly state that the red dashed line represents OR = 1 (no association).

Response 1: We have revised the caption of Figure 1 accordingly (lines 230-233).

Comment 2: Threshold Annotation: Adding a vertical dotted line at TG-HDL ≈ 2.5 with a brief note (e.g., “threshold”) would help readers quickly interpret the inflection point.

Response 2: We have revised Figure 1 accordingly.

Comments on Figure 2:

Comment 1: “The legend lists “TG_HDL” and “TyG_HDL.” However“TyG_HDL” is not described in the text.”

Response 1: We have removed the TyG-HDL from Figure 2.

Comment 2: “Some lines are very close together (AIP, TG-HDL). Consider using dashed/dotted styles or clearer color differentiation.”

Response 2: We have made necessary edits and replaced Fig 2 with a new plot.

Comment 3: “It would be helpful to add AUC values in the legend (e.g., “AIP (AUC = 0.625)”) for immediate comparison.”

Response 3: We have added the AUC values in the new plot.

Comment 4: “BMI” is included in the ROC but not discussed in the predictive performance results. Either explain its inclusion or remove it to avoid distraction.”

Response 4: We have removed the BMI from the new plot.

Reviewer #2

Thank you for taking the time to review our work. We’re grateful for your guidance throughout the review process.

Comment 1: “Authors failed to picture the rationality of this study? What is main novelity? Previous study answer this question how you fill the gap?”

Response 1: The novelty of our study lies in the comprehensive, PCOS-specific evaluation of multiple metabolic indices, including the first assessment of AIP in relation to infertility, along with non-linear modeling and ROC analysis to assess clinical relevance. While we mentioned these in the last paragraph of our Introduction section, we have revised the text (lines 79-89) to more explicitly highlight these gaps and to clarify the novelty of our study.

Comment 2: “How many women you excluded due to the exclusion criteria?”

Response 4: We have clarified the number of participants excluded (n=335) based on the predefined exclusion criteria in the Study Design and Participants section (lines 160-161).

Comment 3: “How were the techniues of assessments , kits, methodes…”

Response 4: We have revised the Measurements and Tools section to provide additional details on the assessment techniques and methods (lines 103-109).

Comment 4: “How was diagnostic criteria of PCOS?”

Response 4: We have clarified the diagnostic criteria of PCOS in the Measurements and Tools section (lines 118-122).

Comment 5: “What is definition of infertility?”

Response 5: We have clarified the definition of infertility in the Measurements and Tools section (lines 110-117).

Comment 6: “It was better to have a control group for comparison.”

Response 6: The primary objective of this study was to examine the association between metabolic indices and infertility within a population of women with PCOS, rather than to compare women with and without PCOS. Accordingly, we used an internal comparison between infertile (cases) and fertile (control) women with PCOS to minimize confounding related to the diagnosis of PCOS itself and to better characterize metabolic heterogeneity within this high-risk group.

While inclusion of a non-PCOS control group may be informative for other research questions, it was beyond the scope of the present study.

Comment 7: “Please discuss about any pollsible bias of study.”

Response 7: We have expanded the Strengths and Limitations section to explicitly discuss potential sources of bias(including selection bias related to the tertiary-care setting, residual confounding from unmeasured lifestyle and reproductive factors, etc.) (lines 344-351).

Comment 8: “What is clinical implication of study?”

Response 8: We have added a Clinical Implications subsection to the Discussion to clarify the potential relevance of our findings (lines 318-327).

Comment 9: “Editor advised to check the usig of AI.”

Response 9: We confirm that no artificial intelligence tools were used for data collection, statistical analysis, result generation, or scientific interpretation in this study. Any language editing, where applicable, was limited to improving grammar and clarity, and all content was reviewed and approved by the authors.

Sincerely,

Dr. Amene Abiri

---

## [Decision Letter · Decision Letter 1]

11 Feb 2026

Association between different atherogenic and insulin resistance indices and infertility in polycystic ovary syndrome patients

PONE-D-25-55841R1

Dear Dr. Abiri,

We’re pleased to inform you that your manuscript has been judged scientifically suitable for publication and will be formally accepted for publication once it meets all outstanding technical requirements.

Kind regards,

Ricardo Ney Oliveira Cobucci, Ph.D

Academic Editor

PLOS One

Additional Editor Comments (optional):

Congratulations.

Reviewers' comments:

Reviewer's Responses to Questions

**Comments to the Author**

1. If the authors have adequately addressed your comments raised in a previous round of review and you feel that this manuscript is now acceptable for publication, you may indicate that here to bypass the “Comments to the Author” section, enter your conflict of interest statement in the “Confidential to Editor” section, and submit your "Accept" recommendation.

Reviewer #1: All comments have been addressed

2. Is the manuscript technically sound, and do the data support the conclusions?

Reviewer #1: Yes

3. Has the statistical analysis been performed appropriately and rigorously? 

Reviewer #1: Yes

4. Have the authors made all data underlying the findings in their manuscript fully available?

Reviewer #1: Yes

5. Is the manuscript presented in an intelligible fashion and written in standard English?

Reviewer #1: Yes

6. Review Comments to the Author

Reviewer #1: (No Response)

7. PLOS authors have the option to publish the peer review history of their article (what does this mean?). If published, this will include your full peer review and any attached files.

Reviewer #1: No

---

## [Editor Report · Acceptance letter]

PONE-D-25-55841R1

PLOS One

Dear Dr. Abiri,

I'm pleased to inform you that your manuscript has been deemed suitable for publication in PLOS One. Congratulations! Your manuscript is now being handed over to our production team.

Kind regards,

on behalf of

PROFESSOR Ricardo Ney Oliveira Cobucci

Academic Editor

PLOS One